# Learning to Generate Object Interactions with Physics-Guided Video Diffusion

## Abstract

Recent models for video generation have achieved remarkable progress and are now deployed in film, social media production, and advertising. Beyond their creative potential, such models also hold promise as world simulators for robotics and embodied decision making. Despite strong advances, however, current approaches still struggle to generate physically plausible object interactions and lack physics-grounded control mechanisms. To address this limitation, we introduce *KineMask*, an approach for physics-guided video generation that enables realistic rigid body control, interactions, and effects. Given a single image and a specified object velocity, our method generates videos with inferred motions and future object interactions. We propose a two-stage training strategy that gradually removes future motion supervision via object masks. Using this strategy we train video diffusion models (VDMs) on synthetic scenes of simple interactions and demonstrate significant improvements of object interactions in real scenes. Furthermore, KineMask integrates low-level motion control with high-level textual conditioning via predictive scene descriptions, leading to effective support for synthesis of complex dynamical phenomena. Extensive experiments show that KineMask achieves strong improvements over recent models of comparable size. Ablation studies further highlight the complementary roles of low- and high-level conditioning in VDMs. Our code, model, and data will be made publicly available.

## 1 Introduction

Recent years have seen substantial advances in video generation, with Video Diffusion Models (VDMs) emerging as a leading paradigm for high-resolution, temporally consistent synthesis (Ho et al., 2020; Blattmann et al., 2023b; Yang et al., 2025; Kong et al., 2024). This progress has elevated visual quality and enabled early commercial use in creative performances (Miller, 2023), experimental filmmaking (Chayka, 2023), and advertising (Roth, 2025). Beyond content creation, VDMs have also been explored as world models (Ha & Schmidhuber, 2018), capable of anticipating real-world interactions and supporting robotics and embodied decision making (Agarwal et al., 2025; Alonso et al., 2024; Ding et al., 2024). Realizing this vision, however, requires strict physical plausibility and control, since small deviations from realistic physics can accumulate into large errors in predicted dynamics. Yet current VDMs struggle to capture fundamental traits (Motamed et al., 2025; Kang et al., 2025) such as object permanence and causal interactions. Hence, methods for physically accurate video generation are a key step toward establishing VDMs as reliable world models.

Recent frameworks such as PhysGen (Liu et al., 2024b) and WonderPlay (Li et al., 2025c) integrate physics-based simulators into data-driven video generation, but they rely on explicit scene reconstruction, a challenging task on its own. Around the time of this work, Force Prompting (Gillman et al., 2025) introduced physics-guided controls into VDMs using simulated training data. While promising, it remains limited in handling rigid-body interactions, often producing unrealistic dynamics, and distorted shapes. Motivated by these limitations, we propose an approach that enables video diffusion models to generate physically realistic object interactions, as a core requirement for robotics and advanced video generation. We focus on two central questions: **(1)** Can a video diffusion model generate *realistic interactions* between objects given initial dynamic conditions, and **(2)** how do data and textual conditioning influence the emergence of causal physical effects in generated videos?

Figure 1: **KineMask results.** We enable object-based control with a novel training strategy. Paired with synthetic data constructed for the task, KineMask enables pretrained diffusion models to synthesize realistic object interactions in real-world input scenes.

To address these questions, we introduce KineMask, a physics-guided framework for generating object interactions and effects in complex scenes (Figure 1). KineMask provides *low-level* kinematic control over parameters such as object direction and speed, while inferring object interactions directly within the VDM. KineMask is trained on simulator-rendered videos that capture not only physically valid dynamics but also explicit object interactions, paired with textual descriptions of the underlying events. We further leverage object masks as guidance, which enables refined control over motion trajectories and improves the model's understanding of object shapes. Beyond low-level control, KineMask integrates *high-level* prompt conditioning through textual descriptions of future scene dynamics. At inference, it predicts dynamics from an input image and enables the generation of complex effects such as glass shattering or liquid spilling. Extensive experiments demonstrate that KineMask not only introduces new control capabilities but also outperforms state-of-the-art models of comparable size, while ablation studies highlight the importance of both the proposed training strategy and the integration of low- and high-level controls.

In summary, we propose the following contributions:

- We introduce KineMask, a mechanism for object motion conditioning in VDMs, based on a novel two-stage training and conditioning encoding.
- We train KineMask on a synthetic dataset of dynamic scenes with simple object interactions and demonstrate generalization of our model to complex interactions in real scenes.
- We combine low-level motion control with high-level text conditioning, yielding significant gains over comparable models in realistic video synthesis.

## 2 RELATED WORKS

**Video Diffusion Models.** Early video diffusion directly extended image generators by inserting temporal layers into denoising U-Nets (Agarwal et al., 2022; Blattmann et al., 2023a; Guo et al., 2024b; Wang et al., 2023; 2025b; Chen et al., 2023; Bar-Tal et al., 2024). Later, video synthesis improved through the usage of Diffusion Transformers (DiTs) (Peebles & Xie, 2023), inheriting scaling properties from native transformer-based architectures. The use of DiTs allowed for higher-resolution videos and stronger visual quality (Yang et al., 2025; Kong et al., 2024; Wan et al., 2025; HaCohen et al., 2024). We build KineMask on CogVideoX (Yang et al., 2025), benefiting from the advantages of DiTs. Beyond creative purposes, diffusion-based video generation is now used for *world modeling*, by synthesizing the possible outcomes of actions in an environment. (Alonso et al., 2024; Valevski et al., 2025) first proposed diffusion-based world models on restricted scenarios. Some like GAIA (Hu et al., 2023; Russell et al., 2025) train at scale on specific domains such as autonomous driving, while Cosmos (Agarwal et al., 2025) uses large-scale data to handle heterogeneous domains. These models still suffer from limited realism in generating object interactions, motivating our study.

**Control for video generation** Significant efforts have been made to extend control on generated videos beyond textual conditioning. On images, ControlNet (Zhang et al., 2023) and similar approaches (Mou et al., 2024; Zhao et al., 2023) proposed plug-and-play control trainable components for dense conditioning such as edges, depth, or human poses. In videos, VideoControlNet (Hu & Xu, 2024) propagated conditions frame by frame using optical flow, while others focused on pose-driven

human video generation (Zhang et al., 2024b) or keyframe-to-video propagation (Li et al., 2024a). More recent works such as SparseCtrl (Guo et al., 2024a) uses just a few keyframes for sketch, depth, or image conditioning, ignoring motion conditions. On motion, DragDiffusion (Shi et al., 2024) and MotionCtrl (Li et al., 2024b) allow interactive editing and trajectory-aware conditioningm. Motion Prompting (Geng et al., 2025) introduces point-track motion prompts, Tora (Zhang et al., 2025) designs trajectory-oriented diffusion transformers. All these require pre-defined trajectories. Some drive motion by directly guiding attentions (Pondaven et al., 2025) or noise (Burgert et al., 2025) using reference videos. Beyond single modalities, Cosmos-Transfer (Abu Alhaija et al., 2025) demonstrates adaptive multimodal conditioning. To our knowledge, KineMask is the first approach designed to generate object interaction by controlling initial object velocity.

**Physics-aware video generation**  The interactions between physical understanding and video generation is a growing research field. A first line of work integrates physical simulations with learning-based techniques. In particular, PhysDreamer (Zhang et al., 2024a) generates oscillatory motions on 3D Gaussians, while DreamPhysics learns physical properties of dynamic 3D Gaussians with video diffusion priors (Huang et al., 2025). On video generation, WonderPlay (Li et al., 2025c) bridges physics solvers and generative video models to synthesize dynamic 3D scenes across diverse physical phenomena. PhysGen (Liu et al., 2024b) applies a similar approach using only 2D information for rigid body interactions. However, the use of a simulator requires significant engineering efforts and limits the flexibility of models. Differently, C-Drag (Li et al., 2025b) uses a LLM for inferring causal motion on output videos, also requiring tracking-based control. Using purely diffusion models, InterDyn (Akkerman et al., 2025) explores the capability of video models to render realistic object dynamics. However, it exploits frame-wise masks of controlling elements, which are typically unavailable at test time. Similarly to us, Li et al. (2025a) explores physics post-training but focuses on gravity effects. The concurrent work Force Prompting (Gillman et al., 2025) explores similar ideas to ours but does not consider object interactions and deploys simpler motion control. We experimentally compare to Force Prompting and demonstrate improved performance.

## 3 KINEMASK

KineMask is designed to synthesize realistic interactions among objects given the image of an initial scene and initial object velocity encoded by the object mask, see Figure 2. Below we introduce preliminaries in Section 3.1, we then describe our conditioning mechanism in Section 3.2 and outline the training and inference procedure in Section 3.3.

### 3.1 PRELIMINARIES

**Video diffusion Models.**  VDMs generate data by reversing the noising process. In training, a clean video $\mathbf{x}_0 \in \mathbb{R}^{F \times H \times W \times C}$ is perturbed into a noisy version $\mathbf{x}_t$ by adding Gaussian noise at a randomly sampled timestep $t$, and the model is optimized to approximate the corresponding reverse transition. At inference, the process is inverted: starting from pure Gaussian noise $\mathbf{x}_T$, the model denoises through intermediate states $\{\mathbf{x}_t\}_{t=1}^{T}$ until it recovers a clean output video $\mathbf{x}_0$ after $T$ steps. To be grounded to real scenes, we use image-to-video (I2V) models, where the video synthesis is conditioned on a reference image $\mathbf{y}$. Formally, we denote by $p_\theta$ the VDM with parameters $\theta$. Conditioned on a high-level text description of the desired output $c$ and a reference image $\mathbf{y}$, the denoising step is:

$$\mathbf{x}_{t-1} \sim p_\theta(\mathbf{x}_{t-1} \mid \mathbf{x}_t, c, \mathbf{y}), \qquad (1)$$

where $\mathbf{x}_{t-1}$ is a tensor defined over the frame dimensions. The training loss minimizes the KL divergence between the true reverse conditional $p$ and the model distribution:

$$\mathcal{L}_{\mathrm{diff}}(\theta; \mathbf{x}_0, \mathbf{y}, c, t) = D_{\mathrm{KL}}\Big(p(\mathbf{x}_{t-1} \mid \mathbf{x}_t, \mathbf{x}_0) \,\big\|\, p_\theta(\mathbf{x}_{t-1} \mid \mathbf{x}_t, c, \mathbf{y})\Big). \qquad (2)$$

In practice, this objective is implemented by a noise prediction task in which the network learns to estimate the Gaussian noise added to $\mathbf{x}_0$.

**ControlNet.**  To allow additional guidance, a ControlNet (Zhang et al., 2023) branch $\psi_\phi$, parameterized by $\phi$, can encode an arbitrary dense control signal $\mathbf{u} \in \mathbb{R}^{F \times H \times W \times D}$ driving the output generation. For more details, please refer to Zhang et al. (2023). The denoising step then becomes:

$$\mathbf{x}_{t-1} \sim p_\theta\big(\mathbf{x}_{t-1} \mid \mathbf{x}_t, c, \mathbf{y}, \psi_\phi(\mathbf{u})\big). \qquad (3)$$

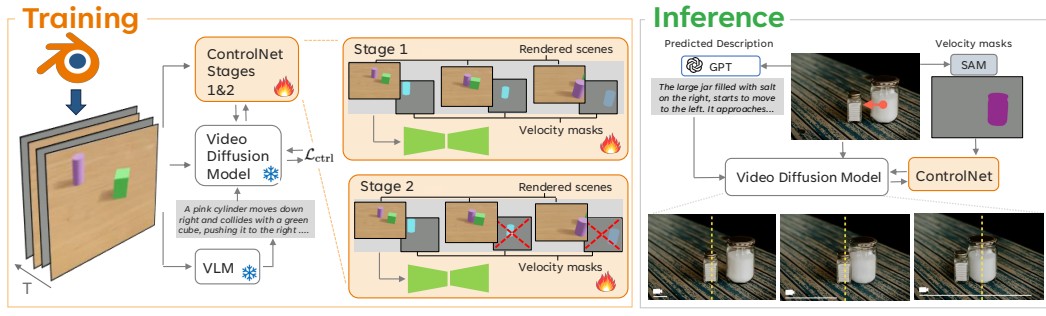

Figure 2: **KineMask pipeline.** We encode our *low-level* control signal as a mask encoding the instantaneous velocity of the moving objects for each frame, to train a ControlNet (left) in two stages using Blender-generated videos of objects in motion. In the first one, we train with all frames, whereas in the second one, we randomly drop part of the final frames. We also provide a *high-level* textual control extracted by a VLM. At inference (right), we construct the low-level conditioning with SAM and use GPT to infer high-level outcomes of object motion from a single frame.

When training the ControlNet, the parameters of the backbone model $\theta$ are kept frozen, while only the control branch $\phi$ is optimized. The corresponding loss is

$$\mathcal{L}_{\text{ctrl}}(\phi; \mathbf{x}_0, \mathbf{y}, \mathbf{u}, c, t) = D_{\text{KL}}\Big(q(\mathbf{x}_{t-1} \mid \mathbf{x}_t, \mathbf{x}_0) \,\big\|\, p_\theta\big(\mathbf{x}_{t-1} \mid \mathbf{x}_t, c, \mathbf{y}, \psi_\phi(\mathbf{u})\big)\Big). \tag{4}$$

### 3.2 Enabling motion control

**First-stage training.** We now want to enable object-wise motion control for KineMask. Specifically, our goal is to move an object in an input scene $\mathbf{y}$ with controlled direction and velocity, allowing us to study the effects of object interactions in the videos generated with diffusion models. To do so, we assume access to a dataset $\mathcal{D}$ of captioned videos depicting objects in motion. Let $f \in \{1, \dots, F\}$ denote the frame index. For each frame $f$ and object in the scene, we are given a mask $\mathbf{m}_f \in \mathbb{R}^{H \times W \times 3}$ aligned with the image resolution. The three channels encode the instantaneous velocity vector, with the red, green, and blue channels corresponding to motion along the $x$-, $y$-, and $z$-axes, respectively, in the pixels defined by a segmentation mask of the object. In this way, $\mathcal{D}$ provides not only spatial information about object locations, but also explicit ground-truth dynamics in three dimensions. The velocity masks are then aggregated into a tensor $\mathbf{m} \in \mathbb{R}^{F \times H \times W \times 3}$ and used to condition $\psi_\phi$. Similarly to Akkerman et al. (2025), *we annotate in this way only the velocity of the objects moving in the first frame of the rendered video*, leaving blank the mask for objects potentially moved by interactions. We visualize our strategy in Figure 2 (top). This enforces the model to synthesize interactions without explicitly relying on pixel control information. We can then train a *first-stage* KineMask ControlNet $\phi'$ by solving

$$\phi' = \arg\min_\phi \;\; \mathbb{E}_{(\mathbf{x}_0, \mathbf{y}, \mathbf{m}, c) \sim \mathcal{D}, t}\Big[\mathcal{L}_{\text{ctrl}}(\phi; \mathbf{x}_0, \mathbf{y}, \mathbf{m}, c, t)\Big], \tag{5}$$

The $\phi'$ network learns to map dense pixel-wise supervision into structured guidance for object motion in the generated videos. We show the training masks in Figure 2 (top).

**Second-stage training.** KineMask $\phi'$ enables motion control given motion masks $\mathbf{m}$ provided for all the frames of a video. While such a setup simplifies training, it does not correspond to our desired scenario of video generation conditioned *only* by the object motion at the first video frame. Towards this goal, we propose a mask dropout strategy, erasing the last part of the velocity masks in $\mathbf{m}$ at training time, as shown in Figure 2 (bottom). Formally, we define a truncated mask tensor

$$\mathbf{m}_\odot = \{\mathbf{m}_{\odot,f} = \mathbf{m}_f \text{ if } f \leq f^*, \; \mathbf{0} \text{ otherwise}\}_{f=0}^F. \tag{6}$$

where $f^*$ denotes the cutoff frame index corresponding to the dropout ratio. Thus, only the first frames contain velocity supervision, while the remainder of the sequence is set to zero. We then

train a *second-stage* KineMask $\phi''$ by finetuning $\phi'$ with this strategy, solving

$$\phi'' = \arg\min_{\phi''} \ \mathbb{E}_{(\mathbf{x}_0,\mathbf{y},\mathbf{m},c),\sim\mathcal{D},t}\Big[\mathcal{L}_{\text{ctrl}}(\phi';\mathbf{x}_0,\mathbf{y},\mathbf{m}_\odot,c,t)\Big]. \tag{7}$$

As a result of the dropout during training, the VDM equipped with $\phi''$ is able to move objects by taking as input only the *initial* velocity, with $\mathbf{m}_\odot = \{\mathbf{m}_0,\mathbf{0},...,\mathbf{0}\}$. Ultimately, to render realistic videos, the VDM *must synthesize motion dynamics* starting from initial conditions only.

### 3.3 DATA

**Training.** We show our training pipeline in Figure 2 (left). For training $\phi''$, we assumed the availability of a dataset $\mathcal{D} = \{(\mathbf{x}_0,\mathbf{y},\mathbf{m},c)\}$. Besides the target video $\mathbf{x}_0$ and the reference conditioning image $\mathbf{y}$, we require both *low-level* and *high-level* conditioning for physical dynamics. At the low level, we require the aggregated velocity masks $\mathbf{m}$ defined in Section 3.2. At the high level, instead, we associate each video with a textual description $c$ summarizing the effects of physical interactions. Since collecting real-world videos with such annotations is impractical, we generate synthetic data in Blender. Importantly, such simulated data still allows to generalize to real scenes, as we empirically verify in our experiments. We render scenes with boxes and cylinders placed on textured surfaces, and assign to each controlled object an initial velocity with random direction and magnitude. This procedure yields $\mathbf{x}_0$ as the rendered video, $\mathbf{y}$ as the first frame of the sequence (used for image-to-video conditioning), and $\mathbf{m}$ as the stack of per-frame velocity masks, which provide supervision of motion. To obtain the high-level descriptions $c$, we instead process each rendered video with a vision–language model (VLM), prompted to provide detailed video captions with particular focus on object interactions. The full prompt used for caption generation is reported in Appendix A.2.

**Inference.** At inference time, we assume as input an unseen image $\mathbf{y}$. An object mask can be easily obtained for the target object e.g., using SAM2 (Ravi et al., 2025), while the desired object velocity at the first frame is assumed to be provided by the user. We use this information to construct $\mathbf{m}_\odot$. We also prompt GPT-5 (OpenAI, 2025) for a description $c_{\text{infer}}$ of the effects on the scene if the object starts moving in the direction indicated by the user. The full prompt is in Appendix A.2. Combining those with random noise $\mathbf{x}_T \sim \mathcal{N}(0,1)$, we construct the input tuple $\{\mathbf{x}_T, \mathbf{y}_{\text{input}}, \mathbf{m}_\odot, c_{\text{infer}}\}$ compatible with our VDM equipped with $\phi''$. Our inference pipeline is illustrated in Figure 2 (right).

## 4 EXPERIMENTS

We now present our experiments and describe the experimental setup in Section 4.1. Section 4.2 next presents our main results and compares KineMask to the state of the art. Finally Section 4.3 presents a comprehensive analysis with ablations for different components of our method. Further ablations are in Appendix A.3.

### 4.1 SETUP

**Datasets.** We generate two datasets for training and evaluation. Following Section 3.3, we render cubes and cylinders with random colors, moving on textured backgrounds from AmbientCG (AmbientCG, 2018-2025). The first *Interactions* dataset contains objects moving in random directions and interacting with each other. We also construct a *Simple Motion* dataset, where isolated objects are moving in random directions without collisions. For both, we generate 10,000 training and 100 test samples. Test videos use disjoint colors and textures compared to training to ensure diversity. We visualize samples of the generated data in Appendix A.2. We also include a *Real World* set of 50 images, collected from the web or generated with ChatGPT's image generator (Hurst et al., 2024), used to assess generalization to real scenes with complex objects. Note that unlike *Simple Motion* and *Interactions*, *Real World* does not include ground-truth motion trajectories. We use Tarsier (Wang et al., 2025a) to extract predictive captions $c$.

**Implementation.** We adopt CogVideoX-I2V-5B (CogVideoX) as backbone for KineMask. We use ControlNet on the first 8 layers of the model, with weight at 0.5 during training. For $\phi''$, we also apply a non-uniform sampling strategy for $f^*$, where frame selection is biased toward earlier frames, as rigid-body interactions occur most frequently at the beginning of the simulated sequences. Further details are provided in Appendix A.2. We generate 49 frames at inference.

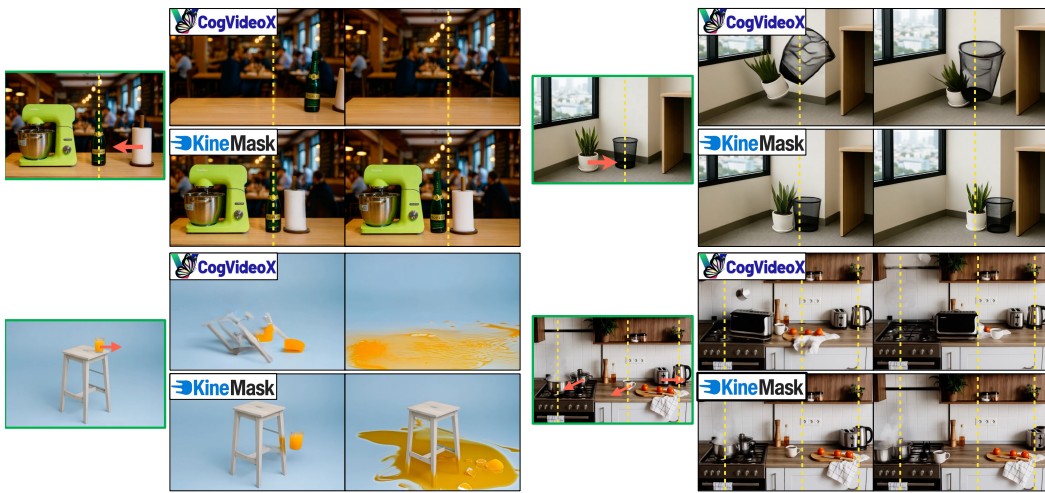

Figure 3: **Qualitative comparison with CogVideoX.** While CogVideoX often suffers from several failure modes, such as hallucinations and incorrect motions, KineMask follows target motion and generates realistic object interactions. In details, we improve object interactions in collisions (top row), show causal effects of object motion (bottom left), and move multiple objects (bottom right).

**Baselines.** We consider two baselines using pretrained image-to-video models: CogVideoX (Yang et al., 2025) and Wan2.2-I2V-5B (Wan et al., 2025) (Wan). We prompt both baselines with the same $c_{infer}$ as used for KineMask. Since we use CogVideoX as a backbone for KineMask, these two methods only differ by our training procedure. Additionally, we evaluate Force Prompting (FP) (Gillman et al., 2025), by mapping our input velocity to input force prompts (Gillman et al., 2025). Force Prompting is also built on CogVideoX. More details on baselines are in Appendix A.2.

**Metrics.** For visual quality, we report the Fréchet Video Distance (FVD) (Unterthiner et al., 2019) and the mean squared error (MSE) between generated and ground-truth videos in our synthetic test sets. For motion, we compute the Fréchet Video Motion Distance (FVMD) (Liu et al., 2024a), which isolates motion quality from appearance. Finally, we use SAM2 (Ravi et al., 2025) to extract semantic masks of objects in both generated and ground-truth videos, and compute Intersection over Union (IoU) between them to assess geometric consistency.

### 4.2 COMPARISON WITH BASELINES

**Qualitative comparison.** We demonstrate the improvements of KineMask against the CogVideoX backbone in Figure 3. We also provide extensive video comparison *on all Real World against all baselines* in the supplementary material (details are in Appendix A.1). We noticed that CogVideoX suffers mainly from hallucinations, making multiple objects fly or completely disappear. Instead, KineMask generates realistic interactions with other objects if they

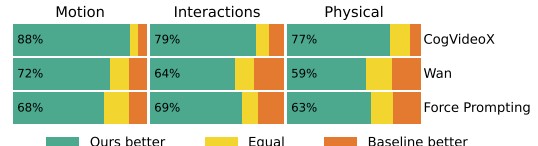

Figure 4: **User study.** We significantly outperform baselines on motion fidelity, interaction quality, and overall physical consistency.

are present in the path of motion of the initial moving object showing a correct understanding of rigid body dynamics. We also show (Figure 3, bottom left) complex interactions that require implicit 3D understanding, making the glass of juice fall and crash as a result of motion, and multi-object motion and interactions (bottom right). We preserve the input motion direction and object consistency in different types of real-world scenarios, showing a strong generalization of the knowledge acquired from simulated videos.

**User study.** In Figure 4, we show results of a user study with 30 participants, who perform pairwise comparisons of videos generated by KineMask and baselines for *Real World*. Participants are presented with initial images and desired object velocities indicated by arrows. We ask participants to evaluate output videos derived from *Real World* on three different axes: motion

(a) Degrees of freedom.

| Method | Simple Motion | | | |
|---|---|---|---|---|
| | MSE↓ | FVD↓ | FMVD↓ | IoU↑ |
| CogVideoX | 158.3 | 601.1 | 1504.6 | 0.051 |
| $\phi''$ only | 86.3 | 288.8 | 201.0 | 0.237 |
| Ours ($\phi' + \phi''$) | **47.2** | **160.3** | **199.8** | **0.367** |
| $\phi'$ w/ masks | 24.9 | 89.7 | 165.6 | 0.684 |

(b) Quantitative results.

Figure 5: **Analysis on low-level motion control**. In Figure (a), we show different controls of KineMask. We can choose different directions, speed, and objects, opening potential for world modeling. In Table (b), we show that our two-stage training strategy allows to considerably boost results and approach the first-stage training exploiting privileged information ($\phi'$ w/ masks).

fidelity to the control signal (Motion), realism of the object interactions (Interactions), overall physical consistency (Physical). Each answer is collected with a three-answer forced choice format, where we compare our outputs with baselines and we ask for user preferences, allowing also to reply "both have the same quality". The full details are presented in Appendix A.2. The collected user preferences indicate that *our method significantly outperforms all baselines in all questions*. This clearly demonstrates the superiority of KineMask in rendering realistic objects interactions.

### 4.3 ANALYSIS AND ABLATION STUDIES

#### 4.3.1 LOW-LEVEL MOTION CONTROL

In our first set of experiments, we evaluate KineMask on low-level motion control. To do so, we train KineMask on *Simple Motion* with a prompt $c_\emptyset$ ="*An object moving on a surface*", hence discarding the rich description $c$ extracted by Tarsier. At inference, we assume $c_{\text{infer}} = c_\emptyset$. Doing so, we isolate the effects of low-level conditioning from those of high-level textual control, allowing for a fair assessment of KineMask as a motion conditioning method.

**Fine-grained control.** In Figure 5a, we present our results on different degrees of freedom of KineMask. Despite being trained on basic synthetic data, KineMask generalizes motion control to complex real-world scenes, coherently with the findings in Gillman et al. (2025). In particular, we show that KineMask achieves disentangled control over different directions, speed, and objects. This allows for a fine-grained evaluation of motion dynamics, precious for world modeling.

**Two-stage training.** We now show the importance of our two-stage training introduced in Section 3.2. We test KineMask on *Simple Motion* only, to study the effects of motion control in isolation. We first report results of CogVideoX prompted with $c$ as a backbone lower bound. Then, we show the performance of KineMask training directly $\phi''$, without first-stage pretraining. As an upper bound, we also evaluate $\phi'$ using ground truth object masks provided for every frame at test time. In Table 5b, we show how our two-stage strategy boosts considerably all metrics compared to training $\phi''$ directly, approaching considerably the upper bound $\phi'$. We noticed that the IoU is very sensitive to minor displacement errors of the moving objects (0.367 vs 0.684), that do not impact the overall quality of motion as captured by the other metrics.

#### 4.3.2 IMPACT OF DATA

We next explore KineMask for generating realistic interactions of rigid bodies. We compare models trained on synthetic samples with and without interactions, and draw conclusions on the capability of VDMs to learn motion. We still use $c_\emptyset$ at both training and inference, as in Section 4.3.1.

**Data influence.** We evaluate KineMask's capability to synthesize object interactions depending on the training data. We first test the model trained on *Simple Motion* and apply it to *Real World* data. We deliberately set input velocities to directions that should generate collisions among objects in the scene (Figure 6, top). The VDM fails to render realistic collisions. However, KineMask trained on the *Interactions* set produces accurate interactions for the same input velocities (Figure 6, bottom). Hence, *KineMask trained on appropriate data allows to render complex object interactions*.

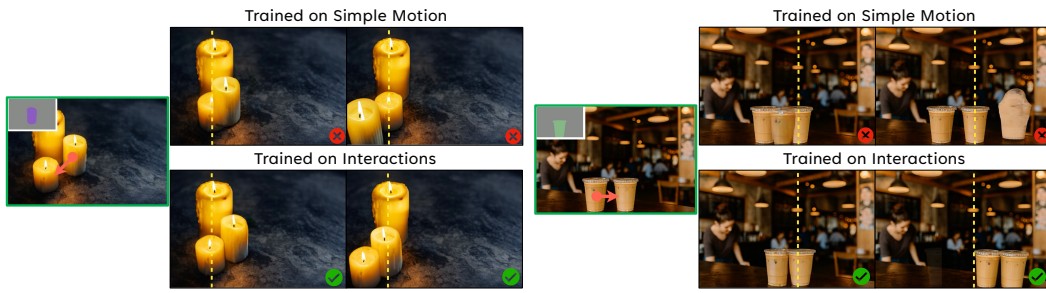

Figure 6: **Impact of training data.** While KineMask trained on *Simple Motion* is able to generalize to *Real World* images, the lack of object interactions in *Simple Motion* results in hallucinations (top). Training on *Interactions* results in collisions and plausible motion of pushed objects (bottom).

Moreover, this experiment yields important insights: while VDMs are robust to visual distribution shift, as even synthetic training data do not influence the realism of *Real World* generated scenes, training on synthetic data with specific motion appears to instead limit the model's generative capabilities. Indeed, the model trained on *Simple Motion* is unable to generate interactions. This raises questions on VDMs catastrophic forgetting.

**Emergence of causality.** Figure 7 shows results of KineMask trained on *Interactions* and tested with three different velocities on a *Real World* scene with interacting objects. As velocity increases, the resulting interactions also change, indicating that the model captures the causal structure of motion. In particular, the final position of the second television varies with the velocity of the first, moving further if the first hits it at higher speeds. This property is valuable for world modeling, as it enables the analysis of different outcomes of object interactions and supports informed planning.

**Quantitative evaluation.** We further validate our findings with a quantitative evaluation on *Interactions* data, following the same protocol as in Section 4.3.1. As shown in Table 1, KineMask achieves the best results when trained on *Interactions*. Notably, even when trained only on *Simple Motion*, it still outperforms all baselines, highlighting that KineMask is the most suitable method for rendering motion give an initial velocity.

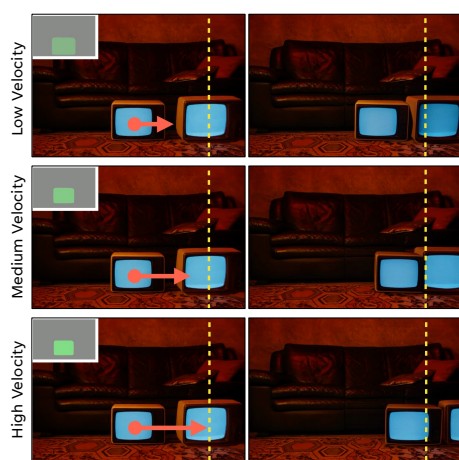

Figure 7: **Causal effects.** The VDM with KineMask correctly identifies causality, with objects having different effects on the scene depending on their initial velocity.

| Method | Training | MSE↓ | FVD↓ | FMVD↓ | IoU↑ |
|---|---|---|---|---|---|
| | | | | *Interactions* | |
| CogVideoX | - | 344.6 | 807.3 | 3514.9 | 0.192 |
| Wan | - | 431.4 | 977.7 | 3649.8 | 0.192 |
| Force Prompt | - | 312.8 | 453.7 | 285.9 | 0.273 |
| KineMask | *Simple Motion* | 166.2 | 301.1 | 160.5 | 0.334 |
| KineMask | *Interactions* | **158.7** | **250.7** | **143.8** | **0.355** |

Table 1: **Effects of data.** Compared with models trained on *Simple Motion*, *Interactions* data considerably boosts performance.

### 4.3.3 HIGH-LEVEL TEXT CONDITIONING

We now enable the usage of $c$ at training and $c_{\text{infer}}$ at inference. This allows us to assess the effects of text introduced in KineMask, and to draw insights on generalization.

**Qualitative evaluation.** We compare KineMask trained with $c_\emptyset$ against training with $c$, while using $c_{\text{infer}}$ for inference in both cases. Doing so, we aim to evaluate the effects of textual prompts describing interactions during training. As shown in Figure 8, training with rich captions $c$ allows the method to render interactions beyond those used for training, successfully exploiting the VDM prior knowledge in the rendered videos. Indeed, the model trained with $c_\emptyset$ fails to follow some effects in $c_{\text{infer}}$, such as vase breaking, or small waves forming. Instead, the model trained with $c$ is able

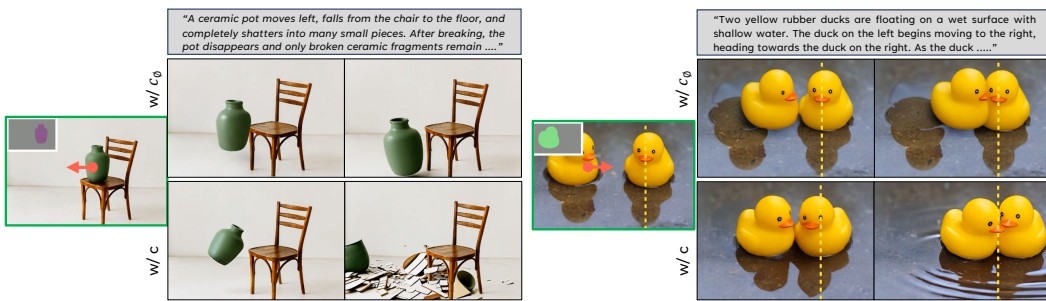

Figure 8: **Impact of text.** Training with rich captions $c$ allows KineMask to generate effects that go beyond the synthetic data used for training, exploiting the prior knowledge of the VDM. Indeed, while we prompt both models with $c_{\text{infer}}$, KineMask trained with $c$ (bottom row) is still able to generate complex effects of the interactions, such as crashing objects, and water effects. See the full prompts in Appendix A.2.

to benefit from additional information described in $c_{\text{infer}}$, correctly breaking the vase and perturbing the water. Please also note that the captions used in training are always tied to synthetic elements in *Interactions*, and therefore have limited diversity. Nevertheless, this still enables successful transfer to $c_{\text{infer}}$ prompts describing complex effects that go beyond the training distribution of KineMask.

**Quantitative evaluation.** We also evaluate metrics with different configurations of textual conditioning in Table 2. We consider trainings with $c_\emptyset$, using $c_\emptyset$ or $c_{\text{infer}}$ at inference. Results demonstrate that using $c$ at training time boosts object consistency (IoU 0.376 vs 0.356 of the second best configuration) and overall realism (FVD 231.3 vs 238.8, MSE on par), while we lose some motion consistency (FMVD 174.4 vs 143.8 of the best model). However, note that Table 2 includes *only evaluation*

| | | Interactions | | | |
|---|---|---|---|---|---|
| **Train** | **Infer** | **MSE ↓** | **FVD ↓** | **FMVD ↓** | **IoU ↑** |
| $c_\emptyset$ | $c_\emptyset$ | **158.7** | 250.7 | **143.8** | 0.355 |
| $c_\emptyset$ | $c_{\text{infer}}$ | 174.4 | 238.8 | 161.3 | 0.356 |
| $c$ | $c_{\text{infer}}$ | 160.9 | **231.3** | 174.4 | **0.376** |

Table 2: **Impact of text.** Training with $c$, we improve object consistency and general quality on synthetic data.

*on synthetic data*, where we have access to a motion ground truth. Considering the improved qualitatives on *Real World* and the benefit on complex interactions (see Figure 8) we believe the use of captions $c$ during training is still beneficial for the final model to preserve its capabilities.

## 5 DISCUSSION

We introduced KineMask, a physics-guided framework that combines low-level motion control with high-level text conditioning to enable video generation of object interactions and effects in complex scenes. Our experiments show that KineMask achieves significant improvements over state-of-the-art models of comparable size, synthesizing realistic multi-object interactions while allowing control over variable object velocities. We believe this methodology can inform future work on world models, with potential implications for robotic manipulation, planning, and other applications of embodied decision making.

While effective, KineMask's low-level conditioning is limited to velocity, whereas real-world motion also depends on factors such as friction, shape, mass, and air resistance. Incorporating such controls is a promising direction for making VDM-generated motion more physically accurate. In our inference pipeline on real images (Section 3.3), we further use ChatGPT to generate textual descriptions of rigid-body interactions. As shown in Section 4.3.3, combining text-based conditioning with KineMask enhances realism, highlighting the complementary roles of high-level textual guidance and low-level video control. These results support the joint use of both modalities for physically grounded world modeling, and we speculate that advances in multimodal language models (Shukor et al., 2025; Team et al., 2023) may enable text-based physical reasoning to complement video generation.

## REPRODUCIBILITY STATEMENT

We currently provide all the details, related to architecture, training stages and hyperparameters in the main paper, Section 4.1, and Appendix A.2. We also include the exact prompts used for the LLMs in Appendix A.2. To further improve reproducibility, will release data, the model, and the source code, along with usage instructions.

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

# A  APPENDIX

## A.1  VIDEO QUALITATIVE RESULTS

For a comprehensive qualitative evaluation on videos, we refer the reader to the attached supplementary material, in which we provide qualitative examples for all *Real World* images animated with KineMask and baselines. We organize the results by using a website that can be easily opened in any browser. To visualize the videos, simply click on the `index.html` file after unzipping the material. Please note that for transparency we include randomly selected outputs from all images in *Real World*. Hence, quality across samples may differ.

## A.2  ADDITIONAL DETAILS

**Used prompts**   We report here the prompt used for $c$ generation at training time and $c_{\text{infer}}$ used for inference time. We prompt Tarsier (Wang et al., 2025a) to extract $c$ with: *"Describe the video in a concise way, covering all motions and collisions between objects."*. This simple prompt is already resulting in sufficiently good descriptions. For inference, instead, we rely on GPT prompting with in-context learning examples to extract suitable descriptions. We report the prompt below, along the associated in-context examples in Figure 9.

---

**Inference-time $c_{\text{infer}}$ generation prompt**

Reference: The first five attached examples are the first frames of simulated videos that contain multiple objects on a surface, in each example one or two objects can start moving, and these objects can or cannot collide with other objects depending on their initial velocity and direction. The range of velocities for the initial motion of these objects is from 0.5m/s to 1.5m/s.

- In the first initial frame the green cube moves at 1.23m/s and the red cylinder at 1.07m/s. The entire video description for this video is: "On a wooden floor, there are four objects: a red cylinder, a green cube, a yellow cube, and a white cylinder. The red cylinder starts in the center and moves to the right, while the green cube moves to the left. The red cylinder continues to move to the right and eventually collides slightly with the white cylinder. The green cube continues to move left and collides slightly with the yellow cube. The red cylinder and the white cylinder remain stationary after the collision.".

- In the second initial frame the white cube moves at 0.67m/s and the purple cube at 1.35m/s, the entire description for this video is: "On a grassy background, there are a yellow cylinder, a purple cube, a pink cube, and a white cube. The white cube moves towards the yellow cylinder and collides slightly with it. The white cube then stops next to the yellow cylinder. The purple cube moves towards the pink cube and collides with it, moving it towards the top right, after the collision all objects remain stationary.".

- In the third initial frame the white cube moves at 1.5m/s and the light blue cube at 1.43m/s, the entire description for this video is: "Four cubes of different colors (yellow, green, blue, and purple) are on a textured, brown surface. The blue cube moves towards the green cube, colliding with it and pushing it downwards. The yellow cube moves towards the pink cube, they collide and the purple cube moves out of the frame. After the collision the blue and green cylinder are static next to each other, while the yellow cube is positioned in the top right part of the frame.".

- In the fourth initial frame, the gray cube moves at 0.64m/s, the entire description for this video is: "The scene is set on a brick ground with patches of green moss. A white cylinder is stationary in the background. A gray cube, initially positioned on the left side of the frame, moves horizontally to the right and eventually stops in the center of the frame. Two pink cubes, one on the right side and another slightly behind it, remain stationary throughout the sequence.".

- In the fifth initial frame the white cylinder moves at 0.71m/s and the pink cylinder moves at 0.55m/s. The entire description for this video is: "The scene consists of a grassy background with four 3D shapes: a purple cube, a white cylinder, a pink cylinder, and another purple cube. The white cylinder and the pink cylinder move slightly to the left. The purple cubes in the foreground remain stationary throughout the sequence.".

Objective: -The last attached initial frame consists of a real picture, in this case the xxx on the xxx moves to the xxx at xxxm/s !!. According to this information and taking into account the provided examples from simulated videos, please provide an entire description, predicting what is going to happen in this scene. The provided video description should follow the same style and words used in the given simulated examples !! , reason about the real scene and describe if collisions happen or not, if they do, describe them, consider the distance and velocity between objects in the real scene to determine how the objects move, and how strong the collisions are. Make sure the description would include realistic effects if they happen produced by the movement or collisions that may occur between objects, for example movement of liquids, steam effects, falling objects, etc. If these effects are not present in the scene please do not include anything related to them in the prompt !!. Do not include velocity values in it, friction effects, information from the surface, inclinations, rotations, or produced sounds.

---

**Training Hyperparameters**   We train KineMask using bf16 mixed precision for 1000 steps with a total batch size of 40 and saved checkpoints every 250 steps. Our backbone is initialized from the pretrained THUDM/CogVideoX-5b-I2V model. For the Controlnet we used the first 8 transformer layers, a downscaling factor of 8 and a control-weight of 0.5. We use AdamW as the optimizer with a learning rate of $1 \times 10^{-4}$, $\beta_1$=0.9, $\beta_2$=0.95, a cosine-with-restart learning-rate schedule and a max gradient norm of 1.0.

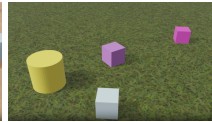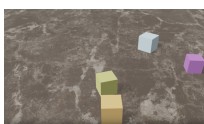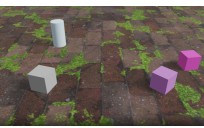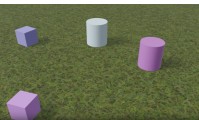

Figure 9: **In-context learning examples.** Providing examples to GPT helps the generation of the text $c_{\text{infer}}$ following a desired format.

**Baseline implementation details**   We compare KineMask with Force Prompting, CogvideoX, and Wan. All methods use an input prompt and an initial image. Additionally, Force Prompting takes the point coordinates of where to apply the force, the force prompt, and the angle of movement. In all cases, we apply the same prompt $c_{\text{infer}}$ that we extract by querying GPT. In the extracted prompts, the moving object and its interactions are always described. Our results demonstrate that for low-level control (CogvideoX and Wan), the simple application of textual conditioning results in ambiguous information, ultimately yielding generation artifacts.

**Synthetic Samples**   Figure 10 shows our synthetic dataset along with the velocity masks for each case. Please note that the masks are extracted only for objects that move at the beginning of the video, these change their color and intensity depending on motion direction and velocity magnitude.

**Prompts for Figure 8**   We report here the prompts used for rendering Figure 8, that we omitted due to the lack of space.

> **Full Prompt Figure 8 (left)**
>
> "A ceramic pot moves left, falls from the chair to the floor, and completely shatters into many small pieces. After breaking, the pot disappears and only broken ceramic fragments remain scattered on the floor."

> **Full Prompt Figure 8 (right)**
>
> "Two yellow rubber ducks are floating on a wet surface with shallow water. The duck on the left begins moving to the right, heading towards the duck on the right. As the duck on the left advances, small ripples spread outward from its base, disturbing the water. The duck on the left continues its motion and collides firmly with the duck on the right. The impact creates overlapping ripples and small splashes around both ducks. The duck on the right shifts to the side, while the duck on the left comes to a stop next to it. After the collision, the water ripples gradually spread and fade, and both ducks remain stationary."

**Details for user study**   We now provide additional details for our user study. First, we divide users into 5 groups, randomizing the videos to display to limit the time necessary to complete our study. We display the videos in couples, along the original frame with the represented control, via a Telegram bot specifically used for the task and represented in Figure 11. For each video pair, we ask:

1. Which of these two outputs follow better the motion indicated by the arrow in terms of direction?
2. Which of the two outputs has the most realistic object interactions?
3. Which of the two outputs have the best overall physical realism?

We aggregate replies from all users in order to build the plots in Figure 4.

### A.3   ADDITIONAL RESULTS

**Velocity encoding**   We propose an ablation on our encoding mechanism. In KineMask, we assume to encode the instantaneous velocity of objects in frame $t$, hence $v_t$. This leads to barely visible masks in the presence of low velocity magnitude when the velocity decreases in presence of interactions. We have tested another configuration, by encoding instead for each frame the initial velocity $v_0$, thus having well-defined masks for all the motion trajectory. However, from the results in Table 3a, we see that encoding the per frame velocity $v_t$ yields better results. We speculate that this strategy would help the model understand the decrease trend of velocity that naturally occurs in motion, ultimately serving as an implicit regularization term for training.

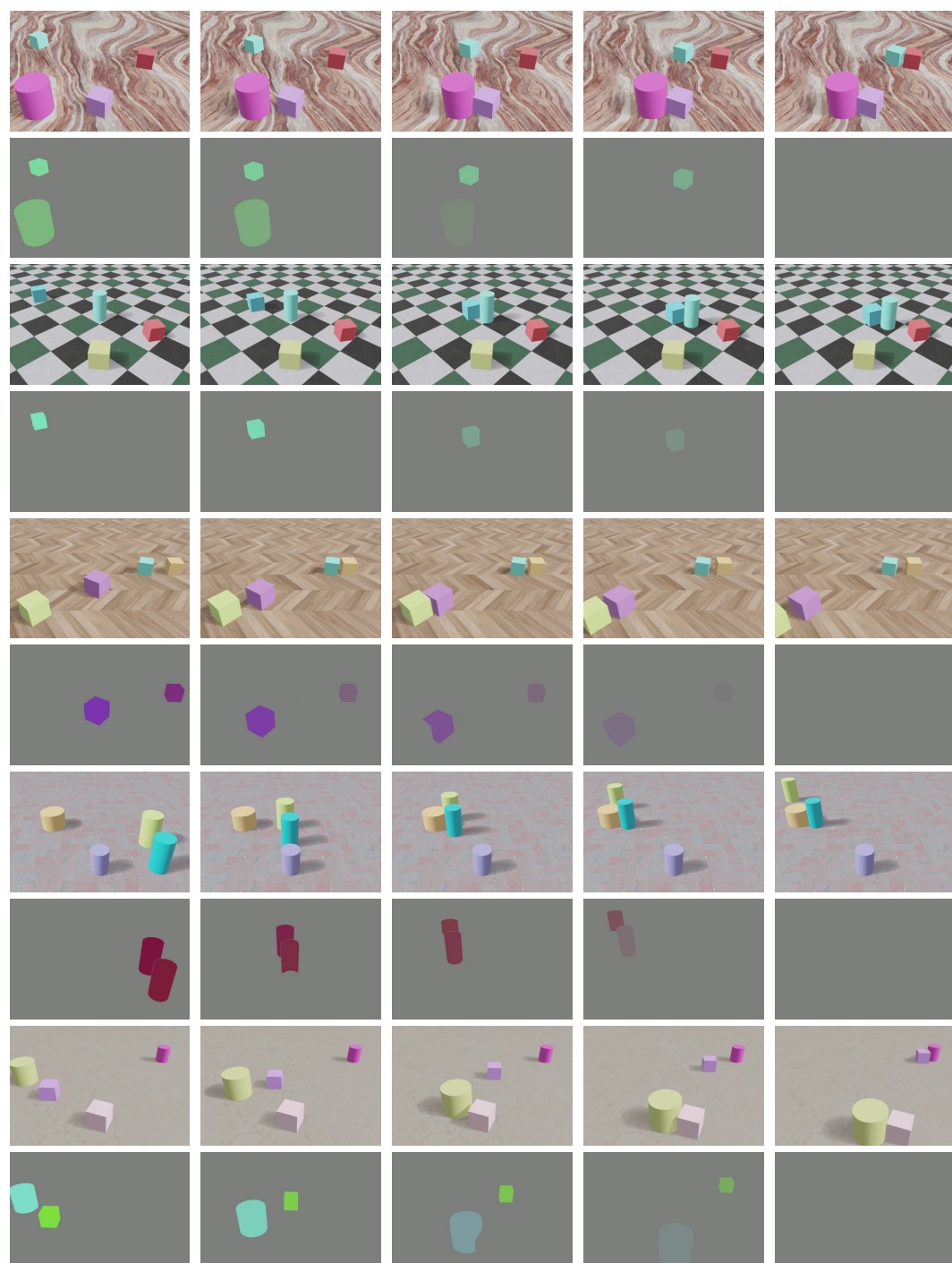

Figure 10: Samples from our synthetic videos and velocity masks used as training dataset.

**Dropout density**  In Section 4.1, we mentioned that we sample non-uniformly the frame $f^*$ for our second-stage training. In Table 3b, we present a quantitative evaluation of the effects of this design choice. As visible, sampling the dropout around the frames where collisions occur in the training set helps KineMask to focus on interaction-rich representations.

**Handling complex objects**  In Figure 12, we report an interesting failure mode of the concurrent Force Prompting succesfully handled by KineMask. As visible, Force Prompting struggles when

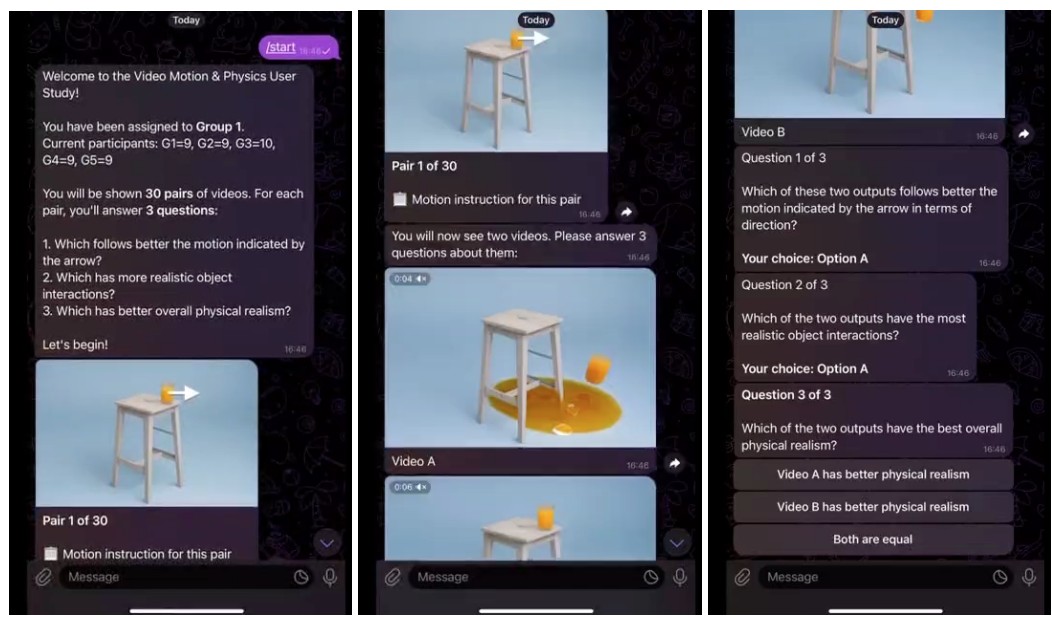

Figure 11: **User study interface.** We use a telegram bot to display pairwise comparisons of videos and baselines, mixing their order. We collect pairwise user preference on direction, interactions realism, and consistency of objects.

| Interactions | | | | |
|---|---|---|---|---|
| **Encoding** | **MSE↓** | **FVD↓** | **FMVD↓** | **IoU↑** |
| $v_0$ | 205.7 | 277.3 | 216.3 | 0.294 |
| $v_t$ (ours) | 160.9 | 231.3 | 174.4 | 0.376 |

(a) Velocity encoding

| | Interactions | | | |
|---|---|---|---|---|
| **Sampling** | **MSE↓** | **FVD↓** | **FMVD↓** | **IoU↑** |
| Uniform | 232.0 | 303.1 | 190.9 | 0.323 |
| Non-uniform (ours) | 160.9 | 231.3 | 174.4 | 0.376 |

(b) Non-uniform sampling

Table 3: **Additional ablation studies.** We report an additional encoding strategies for our conditioning masks in Table (a), proving that our design choice of using $v_t$ is best. Furthermore, we showcase that removing our non-uniform sampling on interactions-rich frames during dropout harms performance (b).

the control is applied to object with thin support structures, or with piled objects. This is the result of an architectural decision, since their control mechanism does not employ masks for identifying the object to move. Instead, KineMask correctly generates motion on these edge cases.

**Failure Cases** Figure 13 shows some failure cases of KineMask, other similar examples can be found in Supplementary Materials. In the first case, we have found that objects that do not have a considerable height tend to ignore others during their motion, thus a collision is not created. The second and third cases, show complex scenarios with many object. This sometimes creates ambiguities, by resulting in object duplication or disappearance. We speculate here that the text prompt may also encourage ambiguity, due to the presence of multiple elements that can be associated to the same textual identifier.

**Qualitatives** For ease of visualization, we report in Figure 14 some additional qualitative results of interactions rendered with KineMask. As visible, in several real scenarios, we are able to render realistic object interactions, where objects move coherently with the users' prompt.

KineMask         Force Prompting

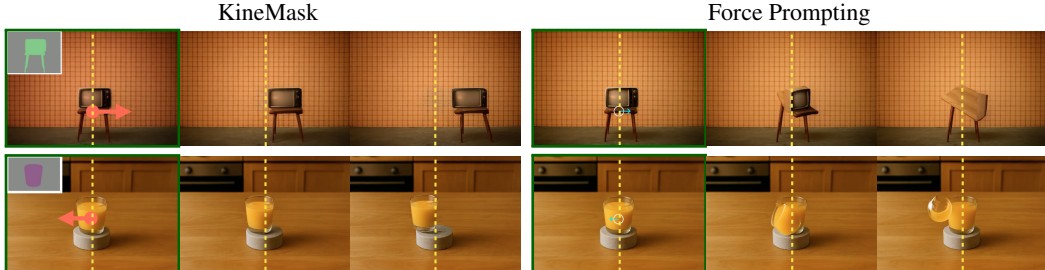

Figure 12: **Motion control comparison.** Our mask-based control is robust to ambiguous scenes where objects overlap, while Force Prompting (the second-best method) suffers from hallucinations (right) due to the ambiguous mapping of the control signal to objects in the scene.

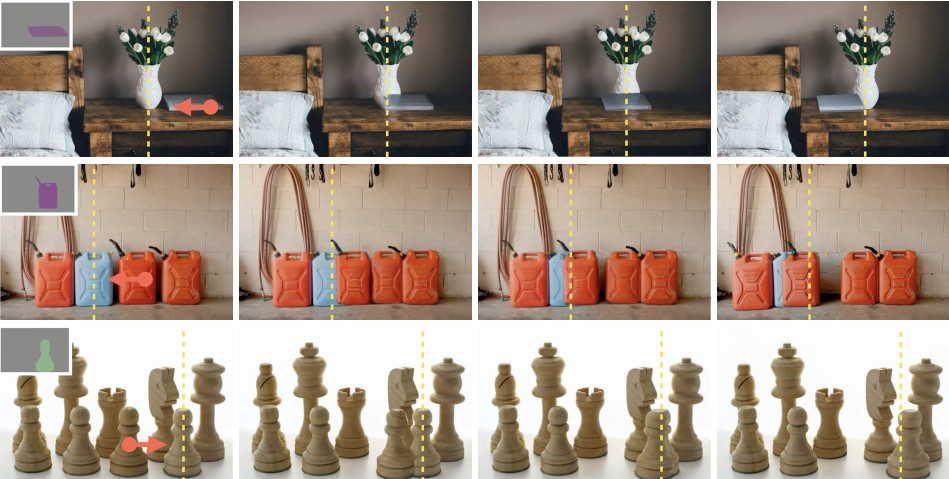

Figure 13: Failure Cases

### A.4 LLM USAGE

In this research, we used an LLM, specifically GPT-5, to aid writing. The usage was limited to polishing some sentences and grammatical checks, as well as to identifying inconsistencies in the used terminologies.

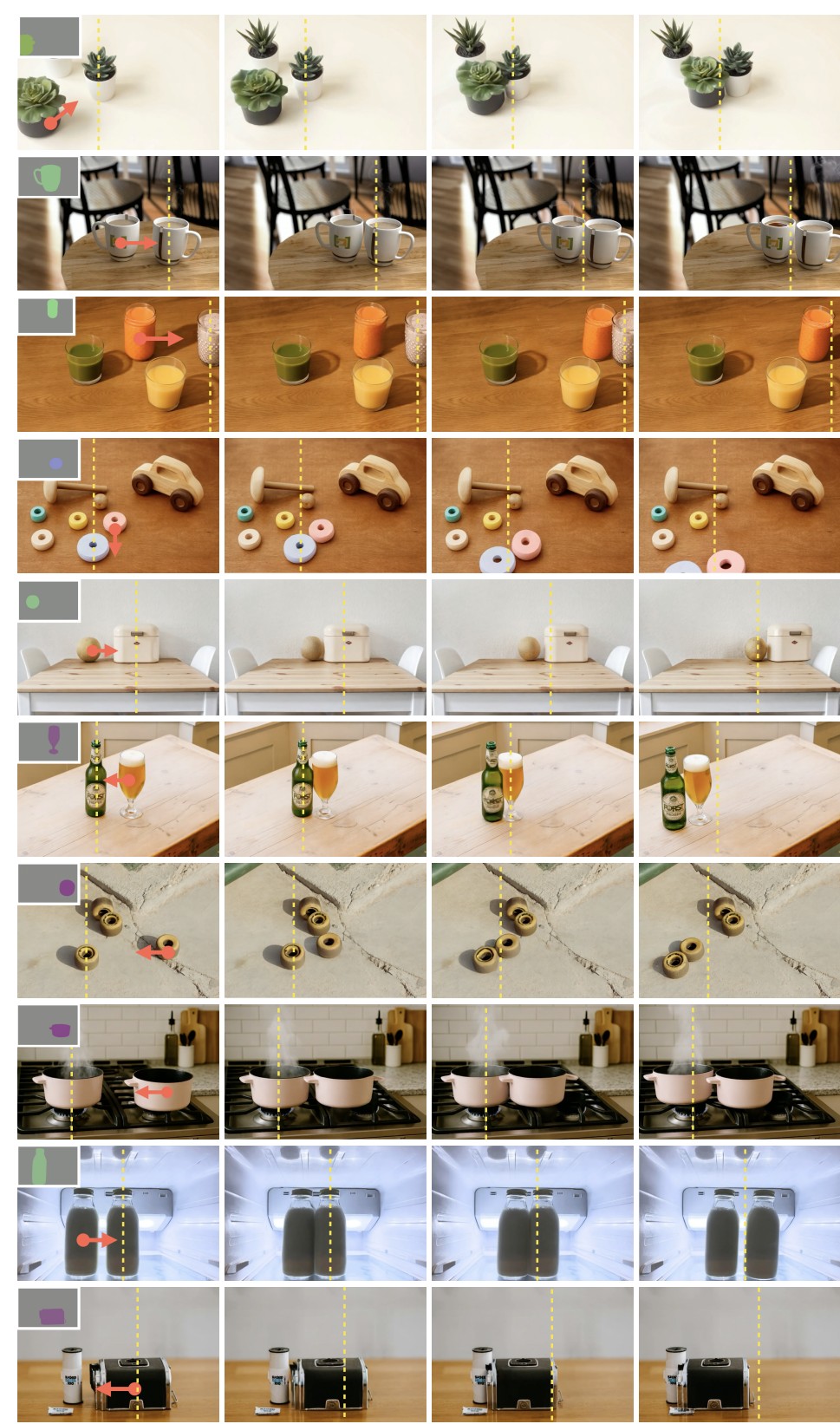

Figure 14: Additional Qualitative Examples of KineMask.

