# OpenReview forum: "Learning to Generate Object Interactions with Physics-Guided Video Diffusion"
_ICLR.cc/2026/Conference — ICLR 2026 Conference Withdrawn Submission_

### Official Review · Reviewer_VAx1 · 2025-10-31

**Soundness:** 3
**Presentation:** 4
**Contribution:** 3
**Rating:** 6
**Confidence:** 3

**Summary:**

This paper introduces KineMask, a novel method for integrating physics-guided control and realistic object interaction into Video Diffusion Models (VDMs). It addresses the key limitation of current video generation models, which often fail to produce physically plausible motion and causal object interactions. A crucial innovation is the two-stage training strategy, consisting of Low-Level Control and High-Level Conditioning. KineMask is trained on simple synthetic scenes (boxes and cylinders), and successfully generalizes to complex interactions in real-world images. It achieves strong improvements over comparable state-of-the-art models in terms of synthesizing realistic motion and object interactions.

**Strengths:**

1. It is interesting that the paper constructs a synthetic dataset of dynamic scenes with simple object (boxes/cylinders) interactions using Blender, and the model trained exclusively on this simple synthetic data can successfully generalize and synthesize complex interactions in real-world scenes.
2. KineMask successfully integrates two distinct levels of control: precise low-level kinematic control (velocity mask) and semantic high-level textual guidance (predicted outcome description). This is also interesting.
3. The core strength is enabling the VDM to infer and generate causal physical interactions (like collisions and liquid spilling) without relying on explicit frame-by-frame guidance.

**Weaknesses:**

1. Training data is limited to basic rigid body interactions. Generalization to complex non-rigid materials (e.g., fire, smoke, cloth) or articulated bodies (e.g., machines) is unproven and likely limited.
2. The use of 3-channel (RGB) velocity encoding in a mask may be a non-intuitive control format for end-users compared to simpler input methods (e.g., a single force vector).

**Questions:**

See Weakness.

---

> ### Author Response · Authors · 2025-11-20
>
> Thanks VAx1 for recognizing the strengths of our work, in particular for noticing the capability of KineMask in generalizing its knowledge obtained from synthetic data to real-world scenarios. Additionally, we appreciate VAx1 for recognizing our main contribution of generating causal physical interactions without relying on explicit guidance. We provide a reply to the weaknesses below.
>
> **1) Limited to Rigid-Bodies:** The overall goal of our work is to learn rigid-body interactions. We explicitly mention in section 5 that this is a limitation of our work. The inclusion of non-rigid materials like fire, smoke, cloth or articulated bodies with extra physical parameters as conditioning is a promising direction for future work.
>
> **2) Input Mask:** Please note that the input for Kinemask is only an arrow which selects the object and indicates velocity, this is the only input needed to be given by a user. It is very simple to map the arrow to a semantic mask and our encoding of velocity using SAM as described in the inference pipeline. We will improve our Figures to clarify this point.

---

### Official Review · Reviewer_HgKF · 2025-10-31

**Soundness:** 1
**Presentation:** 2
**Contribution:** 1
**Rating:** 0
**Confidence:** 4

**Summary:**

This paper constructs a synthetic datasets with belender, containing the ground-truth label of object mask and motion velocity. Then it trains a controlnet to achieve drag-based image-to-video generation.

**Strengths:**

This paper is well written.

**Weaknesses:**

This paper is indeed doing drag-based subject-centric image-to-video generation. And there are many existing works in this track, from Drag-Nvwa, DragVideo to MotionI2V. However, the authors do not discuss about why this paper is claimed to be PHYSICS-GUIDED and do not compare with these existing works. This is my major concern.

Meanwhile, this paper is only trained on simplified rendered data, which makes it hard to generalize to real-world data. As the aforementioned existing papers are trained on Internet data. I wonder the reason of this choice.

**Questions:**

Please refer to the Weaknesses section.

---

> ### Author Response · Authors · 2025-11-20
>
> While we appreciate HgKF’s efforts in reviewing our work, we are saddened to notice that our work got assigned a strong rejection for a misunderstanding, which ultimately led us to withdraw our submission. We hope to clarify here our comparison with drag-based methods, and will explicitly address it for a future version of our paper.
>
> **1.Comparison with drag-based Methods:** We stress that KineMask is **not** a drag-based method for motion control. Indeed, Drag-Nuwa [1], MotionI2V [2], and DragVideo [3] generate motion by assuming to have the full motion trajectory defined as input. While this allows fine-grained controllability, it fully specifies how objects should move for all frames, independently from the environment. This makes it challenging to generate object interactions, since the user would have to imagine realistic motion of the object depending on the interaction. KineMask, instead, accepts only initial velocity conditions as input, and changes the rendered motion depending on the environment, as we show for instance in Figure 7. For a new version of the paper, we will also explicitly mention this difference and quantitatively compare with a drag-based method.
>
> **2. Reason of using synthetic data:** Our physical guidance lies in the usage of masks encoding instantaneous velocity, which is derived by our ad-hoc training data. Please note that this information is not available for internet data, hence the reason for which we decided to generate a dataset with Blender. We will clarify.
>
> [1] Shengming Yin, Chenfei Wu, Jian Liang, Jie Shi, Houqiang Li, Gong Ming, and Nan Duan. Dragnuwa: Fine-grained control in video generation by integrating text, image, and trajectory. arXiv preprint arXiv:2308.08089, 202
>
> [2] Shi, Xiaoyu, et al. "Motion-i2v: Consistent and controllable image-to-video generation with explicit motion modeling." ACM SIGGRAPH 2024 Conference Papers. 2024.
>
> [3] Yufan Deng, Ruida Wang, Yuhao Zhang, Yu-Wing Tai, and Chi-Keung Tang. DragVideo: Interactive Drag-style Video Editing, 2024.

---

### Official Review · Reviewer_ooPX · 2025-10-31

**Soundness:** 2
**Presentation:** 3
**Contribution:** 2
**Rating:** 2
**Confidence:** 5

**Summary:**

The paper proposes KineMask, a method for enabling motion control in pretrained video diffusion models. KineMask trains ControlNet conditioned on velocity masks in two stages. In the first stage the per-frame condition signal is provided for all frames, while the second stage introduces truncation mechanism in the velocity masks to encourage learning causal motion patterns in object interaction scenarios. While the training is done on two synthetic datasets with varying motion complexity, the method successfully leverages video model's priors to generalize to real world scenes.

**Strengths:**

The paper is well-written, easy to follow, and presents the method clearly. The problem of controllable and physically plausible video generation is relevant and important. KineMask is a simple and intuitive approach that tackles this problem. The paper also studies the impact of using combined low-level (motion) and high-level (prompt) conditioning. The visual results look good.

**Weaknesses:**

1. The novelty of the method is quite limited. In essense, KineMask trains ControlNet over a pretrained video diffusion model. The idea of not providing conditioning for the caused motion to encourage learning object interactions is not new (e.g. see [1, 2, 3]). The method thus appears to be an off-the-shelf composition of existing approaches and models (ControlNet, SAM2, GPT-5, Tarsier). Therefore, this hinders the significance of the contribution to the video generation research.

2. The evaluation is also limited to comparisons to only text-controlled models and a single motion-conditioned competitor (Force Prompting). Adding comparisons to other baselines, including ablating different nature of motion condition would definitely strengthen the paper's claims of improved controllability.

3. What drives the learning of object interactions remains largely unstudied. While the second stage of the training and the dropout of conditioning are clearly playing the crucial role in this, the evaluation of dropout ratio is limited to Table 3b in the supplemetary. In general, I would suggest to move the results in Section A3 to the main paper and shift the focus to studying design choices that influence learning of object dynamics and interactions, such as where and how the velocity masks are provided.

4. The quantitative controllability evaluation is limited to synthetic data. However, the claims of emergent causality can also be tested quantitatively on real data. For instance, one could vary the velocity in a certain range and check if the magnitude of the velocity indeed correlates with the magnitude of the generated motion, e.g. measured by optical flow.

5. As also acknowledged by the authors the current design of the conditioning signal is limited to describing rigid translation.

[1] Blattmann, Andreas, et al. "ipoke: Poking a still image for controlled stochastic video synthesis." Proceedings of the IEEE/CVF International Conference on Computer Vision. 2021.

[2] Davtyan, Aram, and Paolo Favaro. "Learn the force we can: Enabling sparse motion control in multi-object video generation." Proceedings of the AAAI Conference on Artificial Intelligence. Vol. 38. No. 10. 2024.

[3] Shi, Xiaoyu, et al. "Motion-i2v: Consistent and controllable image-to-video generation with explicit motion modeling." ACM SIGGRAPH 2024 Conference Papers. 2024.

[4] Zhou, Haitao, et al. "Trackgo: A flexible and efficient method for controllable video generation." Proceedings of the AAAI Conference on Artificial Intelligence. Vol. 39. No. 10. 2025.

[5] Lei, Guojun, et al. "Animateanything: Consistent and controllable animation for video generation." Proceedings of the Computer Vision and Pattern Recognition Conference. 2025.

[6] Chen, Tsai-Shien, et al. "Motion-conditioned diffusion model for controllable video synthesis." arXiv preprint arXiv:2304.14404 (2023).

**Questions:**

1. It is not clear why the authors opt for providing dense velocity masks rather than sparse velocity vectors. The latter would be more flexible and expressive in describing different kinds of motion. Velocity vectors were explored in prior work (e.g. [1, 2, 4, 5, 6] and the drag-based methods acknowledged by the authors in Section 2. Moreover, there exist approaches to cast sparse velocity vectors to dense masks or optical flow [3, 5, 6].

2. It would be interesting to see if KineMask generalizes to unseen controls at test time. For instance, what would happen if the test velocity masks describe non-rigid or non-translation motion.

---

> ### Author Response · Authors · 2025-11-20
>
> We appreciate ooPX’s review and their support for the importance of the task, quality of our manuscript, and focus on our high-level and low-level conditioning mechanism. We reply to their concerns below:
>
> **1. Novelty:** We believe our study is novel, and helps to understand best practices for training diffusion models for interaction generation. Although we use well known components, we believe this should not be ground for rejection, as even very influential papers did not introduce any new architectural element [4].
>
> **2. Regarding comparisons:** We believe our setup is different and novel. [1, 2] map input kinematics to future frames, but require specific architectures and training data, while we repurpose a large-scale video generator. This enables world modeling applications with pre-trained models. Additionally [1] ipoke does not focus on object interactions.  [3] motion-i2v focuses only on motion control, and its mask-based approach hinders the possibility to render realistic interactions. For a new version of the paper, we will also explicitly mention these differences and compare with additional motion-conditioned methods.
>
> **3. Study on Interactions Learning:** Thanks for the suggestion, we have put the dropout ablation study in supplementary for space reasons.
>
> **4. Limitation:** The overall goal of our work is to learn rigid-body interactions. We explicitly mention in section 5 that this is a limitation of our work. The inclusion of other type of motion with extra physical parameters as conditioning is a promising direction for future work.
>
>
>
>
>
> **Questions:**
>
> **Reason of using masks:** Using dense velocity masks gives pixel-level precision and controllability, while using drag based approaches can be ambiguous for object selection in many complex scenes and object setups as shown in Figure 12. In our experiments, we have seen that using dense masks helps the model to improve object consistency. On the other hand, in most of the cited works, while using sparse velocity vectors can be more expressive in some cases, these works require the definition of the full path of motion, which is different from the goal of KineMask where we aim to make the model understand what the path of motion should be and when interactions should happen given only initial dynamic conditions.
>
> **Additional tests:** Thanks for the suggestions, but please note that our work focuses specifically on rigid-body interactions.
>
> [1] Blattmann, Andreas, et al. "ipoke: Poking a still image for controlled stochastic video synthesis." Proceedings of the IEEE/CVF International Conference on Computer Vision. 2021.
>
> [2] Davtyan, Aram, and Paolo Favaro. "Learn the force we can: Enabling sparse motion control in multi-object video generation." Proceedings of the AAAI Conference on Artificial Intelligence.
>
> [3] Shi, Xiaoyu, et al. "Motion-i2v: Consistent and controllable image-to-video generation with explicit motion modeling." ACM SIGGRAPH 2024 Conference Papers. 2024.
>
> [4] Rombach, Robin, et al.
> High-Resolution Image Synthesis with Latent Diffusion Models. In Proceedings of the IEEE/CVF Conference on Computer Vision and Pattern Recognition (CVPR), 2022.

---

### Official Review · Reviewer_eW3K · 2025-11-01

**Soundness:** 3
**Presentation:** 2
**Contribution:** 2
**Rating:** 4
**Confidence:** 5

**Summary:**

This paper introduces KineMask, a framework for guiding pre-trained video diffusion models (VDMs) to generate physically plausible object interactions from a single image. The core problem it addresses is that VDMs, despite their visual quality, lack physical grounding and controllable dynamics.

**Strengths:**

1. The two-stage training with mask dropout is an appropriate way to force the model to learn to infer physics rather than just copy it, enabling it to generate entire interaction sequences from a single initial condition.
2. The combination of a low-level, precise kinematic signal (the velocity mask) with high-level, descriptive text is effective. This allows the model to generate specific, user-defined motions while leveraging the VDM's vast prior knowledge.
3.  The generated interactions are significantly more physically plausible than those from baseline models.

**Weaknesses:**

1. The control is purely kinematic (based on initial velocity). The model does not explicitly account for true physical dynamics like mass, friction, or material properties. It learns plausible visual consequences of motion rather than simulating the underlying physics. In addition, the title uses the term "Physics-Guided," which might imply the use of a physics engine during inference. In reality, the physics is learned implicitly from simulator-generated data during training.
2. The model's ability to generate interactions is shown to be dependent on being trained on a dataset containing such interactions. Its ability to produce physically novel types of interactions not seen during training is likely limited.
3. The demos in Suppl. mostly show the translation of objects and collisions between them, with relatively simple motion types. Furthermore, some samples exhibited issues such as object disappearance and duplication.
4. The proposed method should compare with more methods, such as the video generation model Google Veo 3, and the recent method PhysGen3D[1].

[1] PhysGen3D: Crafting a Miniature Interactive World from a Single Image. CVPR 2025.

**Questions:**

Please see Weaknesses.

---

> ### Author Response · Authors · 2025-11-20
>
> We thank eW3K for the feedback, and for noticing the capability of KineMask of conditioning the motion based on initial conditions only. Moreover, we are glad to see that eW3K appreciated the capability of KineMask to exploit text, and our good results with respect to the baselines. We provide a reply to the weaknesses below.
>
> **1. Control of other physical variables.** We believe there is a misunderstanding. In Section 5, we explicitly mention that this work is limited to rigid-body interactions and velocity as low-level conditioning and that real-world dynamics also depend on factors such as friction, shape, mass and air resistance. While we do not explicitly control those, KineMask still renders realistic motion and interactions, suggesting some degree of physical awareness of such factors, implicitly modeled by our learning strategy. However, we recognize (as stated in Section 5) that this is a promising extension for future work.
>
> **2. Generalization to novel interactions.** Note that, although we train on simple collisions rendered in a synthetic dataset, in Figure 3 and 8, we show complex interactions that require implicit 3D understanding, such as making a glass or a pot fall and crash as a result of motion. As shown in Section 4.3.3, the introduction of descriptive captions at training time allows to preserve the generalization capabilities of the diffusion model to more complex types of motion. We will clarify this in a revised version of the paper.
>
> **3.Simple motion types.** Our work focuses specifically on object interactions for world modeling. We believe that our findings are still valuable for future research, and shed light on the behavior of diffusion models for physically-realistic video renderings. On the samples in supplementary, please note that for transparency we included *all* videos in our Real World dataset. This explains some of the failures. We will clarify.
>
> **4.Comparison with Veo and PhysGen3D.**	Being KineMask a motion conditioning method, we believe it is fair to compare it with alternative conditioning types on models of similar size. Moreover, please note that Veo3 is closed-source, there are no details on its architecture, and it requires significant expenses even for inference, limiting the possibility to establish a fair comparison. However, we would like to point out that, as shown in [1], one of the main limitations of Veo-3 is the lack of low-level controllability as well as the poor understanding of realistic physical interactions, like rigid-body collisions. We will cite this work. On the other hand, PhysGen3D [2] is a similar work to PhysGen [3] (cited in our work). These models work with a simulator in the loop, therefore the motion and interactions are not directly inferred by the video model, which makes it not directly comparable with our work. Indeed, in KineMask the model is the one inferring all the motions and collisions from initial conditions. Please also note that it would be extrenely challenging for PhysGen3D to render complex interactions as those shown in Section 4.3.3, due to the presence of the simulator. We plan to add qualitative comparisons with PhysGen3D in a revised version of the paper.
>
> [1] Thaddaus Wiedemer, Yuxuan Li, Paul Vicol, Shixiang Shane Gu, Nick Matarese, Kevin Swersky, Been Kim, Priyank Jaini, and Robert Geirhos. Video models are zero-shot learners and reasoners. arXiv preprint arXiv:2509.20328, 2025
>
> [2] Boyuan Chen, Hanxiao Jiang, Shaowei Liu, Saurabh Gupta, Yunzhu Li, Hao Zhao, and Shenlong Wang. Physgen3d:Crafting a miniature interactive world from a single image.569 In CVPR, 2025.
>
> [3] Shaowei Liu, Zhongzheng Ren, Saurabh Gupta, and Shenlong Wang. Physgen: Rigid-body physics-grounded image-to-video generation. In ECCV, 2024

---

### Note · Authors · 2025-11-20

**Comment:**

We sincerely thank the reviewers for their time and thoughtful assessment of our work. We would like to address some general points that emerged across the reviews. It seems that there was a misunderstanding on the main goal of KineMask for some of the reviewers. Several comments requested additional comparisons with drag-based methods that work on different scenarios from those addressed by KineMask and focus mostly on explicit motion control with frame-by-frame specification of motion paths. In contrast, KineMask aims to synthesize realistic rigid-body interactions from initial dynamic conditions, without requiring any explicit motion guidance during inference.

We greatly appreciate the reviewers’ efforts and feedback. However, due to the noted misunderstanding, we believe that the requested comparisons and clarifications necessitate additional modifications, for which we decided to withdraw our submission, and prepare a new version of our paper.

**Withdrawal Confirmation:**

I have read and agree with the venue's withdrawal policy on behalf of myself and my co-authors.